# Social Representations of Drinking Water in Schoolchildren and Parents from Two Schools in Zapopan, Mexico

**DOI:** 10.3390/nu13061871

**Published:** 2021-05-30

**Authors:** Alejandra María Corona-Romero, María Fernanda Bernal-Orozco, Gabriela Alejandra Grover-Baltazar, Barbara Vizmanos

**Affiliations:** 1Doctorado en Ciencias de la Salud Pública, Centro Universitario de Ciencias de la Salud (CUCS), Universidad de Guadalajara (UdeG), Sierra Mojada 950, edificio N, Colonia Independencia, Guadalajara ZC 44340, Jalisco, Mexico; lnalejandra.coronaromero@gmail.com (A.M.C.-R.); fera_18@yahoo.com.mx (M.F.B.-O.); grover_1905@hotmail.com (G.A.G.-B.); 2Doctorado en Ciencias de la Nutrición Traslacional, Centro Universitario de Ciencias de la Salud (CUCS), Universidad de Guadalajara (UdeG), Juan Díaz Covarrubias y Salvador Quevedo y Zubieta, edificio C, Colonia Independencia, Guadalajara ZC 44340, Jalisco, Mexico; 3Instituto de Nutrigenética y Nutrigenómica Traslacional, Centro Universitario de Ciencias de la Salud (CUCS), Universidad de Guadalajara (UdeG), Sierra Mojada 950, edificio “P”, Colonia Independencia, Guadalajara ZC 44340, Jalisco, Mexico

**Keywords:** drinking-water, children, social representations, sweetened beverages, behavior, qualitative research

## Abstract

Childhood obesity and children being overweight has increased recently; although they are multi-causal problems, an unhealthy diet is a critical component. In Mexico, drinking water consumption in children from 9 to 18 years only reaches 30% of total fluid consumption. The aim of our study was to describe the social representations (SR) of drinking water in school-children and parents of two schools in Zapopan, Mexico. Associative free listing was used as an information gathering technique. Schoolchildren aged 8 to 12 years (*n* = 50) and parents (*n* = 23) from two elementary schools were selected by a convenience sampling from April to June 2015. A similarity analysis was performed using the co-occurrence index; with this, a similarity graph was obtained. Prototypical analysis was performed to explore the structure of the SR. Three dimensions were described in the children’s SR: a functional dimension related to health and nutrition, a practical dimension that describes the instruments used for its consumption, and a theoretical dimension that specifies the characteristics of water and its relationship with nature. In the parents’ SR, a functional dimension was also found; another dimension was described regarding the integral well-being that drinking water provides. A practical dimension describes the features related to its consumption. The investigation describes the structure of the water SR, which help to contextualize and explain the actions of schoolchildren and their parents regarding drinking water consumption.

## 1. Introduction

The prevalence of children being overweight and childhood obesity has increased in recent decades, positioning itself as a global public health problem. In Mexico, the combined prevalence of being overweight and obesity in children ages 5 to 11 is 33.2% [1]. Although being overweight and obesity are multi-causal problems, an unhealthy diet is recognized as a central axis, and part of those modifiable habits related to non-communicable diseases. The concept of an unhealthy diet includes excessive consumption of sugar-sweetened beverages (SSB), which, according to the guideline of sugar ingestion for adults and children of the World Health Organization, is closely related to the increase in body weight and, therefore, to being overweight and obesity [2].

According to this problem, in Mexico, only 30% of the total fluid intake of children from 9 to 18 years corresponds to drinking water [3]. On the other hand, intake of SSB in Mexico, such as soda and *agua fresca* (which is a non-alcoholic drink made from water, fruit, or cereals, such as oats and rice, as well as added sugar) [4], constitute almost 50% of total fluid intake [5]. Water consumption is related to a lower body mass index [6,7], and is considered the ideal liquid for hydration in children [8]. Therefore, increasing water consumption in children must be a priority in public health action in Mexico.

Tumilowicz, Neufeld, and Pelto [9] refer that for the design, implementation, and evaluation of effective nutrition interventions, it is essential to know the environment of the population and their perspective of the situation. Thus, to increase water consumption in a child population through any intervention, it is necessary to know its cultural environment built by the meaning of the problem and its elements and the interaction between them [10]. In Mexico, interventions based on these cultural elements have proven to be more effective in increasing drinking water consumption in children. [11,12] Social representations (SR) are a global vision of the subjects, or a shared and socially elaborated knowledge regarding an object, which starts from their reality (from their beliefs and experiences, for instance) and which gives meaning and directs behavior, depending on the historical, social, and ideological context of the subjects [10]. They are composed of elements such as beliefs, attitudes, and experiences, among others [13]. In turn, SR is constituted by two elements: their content and their organization. According to the structuralist approach [14], an SR is organized by a central core, determining the organization and meaning of representation. In the central core, it is possible to observe the ideological elements of the representation and the general idea of the object’s functionality. In addition to the central core, the SR structure is made up of peripheral elements that make it possible to know its operation and act within a given social context. Furthermore, it functions as a defense system for the central core against changes in the social context [15,16], protecting it by reinforcing the concepts.

Accordingly, exploring the environment of a population through the SR of drinking water helps describe the subjects’ reality. Thus, the present study aimed was to describe the SR of “drinking water” in schoolchildren and parents of two schools from Zapopan, in Jalisco, Mexico.

## 2. Materials and Methods

### 2.1. Study Design

A qualitative study was carried out based on the Social Representations Theory (SRT) with a structuralist approach [16]. The information was collected from April to June 2015 as part of the “*Gotita de agua* (Water droplet) project”, aiming to design, implement, and evaluate the impact of an educational intervention to increase drinking water consumption in schoolchildren. This project is part of a mixed methodology study [17], consisting of three phases (phase 1, a qualitative study; phase 2, the design of the intervention; and phase 3, implementation and evaluation of the intervention). Phase 1 of this macro project is described in this paper.

The present study adheres to national [18] and international [19] ethical standards. Likewise, informed consent was requested from the parents or guardians of the children, and the verbal assent of the minors was also requested [18].

### 2.2. Study Population, Context, and Recruitment Process

Participating elementary schools were randomly selected; the inclusion criteria were that they belonged to the Jalisco Public Education Secretariat, located in the Guadalajara metropolitan area, that they were coeducational, and that their socioeconomic status was working class. One of the researchers made the invitation in person to the schools that met the criteria. Approximately ten public elementary schools that met these characteristics were selected; however, only one participated. Concerning the private elementary school, elementary schools close to the public school area were selected to characterize these two contexts. In total, six schools were located that met these characteristics; only one agreed to participate.

### 2.3. Data Collection

The selection of the study population was made by a convenience sampling since only children and parents from schools that have agreed to participate were invited to participate [20].

Parents/guardians of the participating children were invited to participate and those who agreed were included in the study. The invitation was made by sending a packet (booklet) with the student’s support, which included a letter of invitation to participate in the study, the procedures and activities to be carried out at the school. It included the letter of informed consent for participation and the data collection form (free lists) to be answered by the parents, with instructions.

About the schoolchildren, we included students from third to fifth grade, ages between 9 and 12, who voluntarily and verbally agreed to participate in the study and whose parents had signed the informed consent sent to them.

For data collection, free listings (a spontaneous evocation of five words from an inducing term) were used as an associative technique [21]. In the present study, the inductive concept was *agua natural,* which refers to “drinking water” (not tap water, since in Mexico it is not safe for consumption). To preserve the spontaneity of the child’s evocation, the words were written down by the evaluator, in the order that the interviewed child mentioned them. The interviewed student was asked to explain why those five words were mentioned [21]. The same evaluator recorded the responses and captured them in a spreadsheet to subsequently carry out data analysis. The information collection tool was applied individually for each student. For the parents’ collection of information, a package was previously sent to them that contained the same information-gathering tool used in children, as well as instructions for its correct completion. All tools that were answered and returned were included in the analysis.

### 2.4. Data Analysis

Similarity analysis was carried out through the co-occurrence index with the words mentioned by the participants, (indicated in italics in the text and bold in the figures). This analysis calculates the number of times that the elements have been mentioned by the participants in the same place (order of evocation) and allows knowing the consensus of the elements mentioned by the informants. A graphic representation of the “drinking water” (similitude graphic) was designed from this index [21,22].

A prototypical analysis was performed in a second stage based on the terms mentioned by the participants in the free listings. The prototypical analysis allows identifying the structure of the SR: the central core and the peripheral elements [16,22,23]. Two elements are weighted in this analysis: the first one evaluates the order of appearance (evocation rank), and the second, the number of times the word had been mentioned among the informants (frequency). A lower rank of appearance refers to the item being cited in the first places of mention among the participants, suggesting that they are the most representative words concerning the SR concept. The crossing of frequency and evocation rank creates four zones that organize the structure of the SR. The first zone is known as the central core, formed by those words with a high frequency of mention and a low order of mention. In the second zone (contrast zone), those elements with low frequency of mention and a low rank of occurrence. Finally, in the peripheral zones, the words have not been mentioned in the first places, but are relevant because of their frequency; these are located in the first peripheral zone, while the second peripheral zone contains words with low frequency and a high occurrence rank [16,21]. These peripheral zones build up the SR body and represent a potential zone of change [14,18]. The data was analyzed in IRaMuteQ software version 07, Alpha 2.

## 3. Results

Fifty students were included. The average age was 10.4 years (SD 1.0); 24 were women, and half of the students came from the private school. On the other hand, 23 parents participated, most of them female (*n* = 21), with an average age of 41.5 years (SD 5.3) (9 parents were from the private school).

### 3.1. Social Representation of Children from Two Schools in Zapopan, Mexico

From the free listing’s associative technique, a total of 247 words or expressions were obtained, of which 147 (59.5%) were similar (degree of consensus of the population regarding the topic).

The analysis of the similarity of the SR of schoolchildren, represented in Figure 1, expresses five groupings of the words mentioned by the population according to the co-occurrence index, reflecting the consensus of the population regarding drinking water. The word “health” was the central element of the SR structure and had the highest co-occurrence index in addition to connecting elements such as “hydration” and “glass”. The words were organized as follows: “health”, the elements “delicious”, “energy”, “white”, “nutrition”, “refreshing”, and “fruit water” (a type of natural fruit-sweetened beverage called *agua fresca de fruta*). The word “hydration”, in turn, connects to “drinking water”.

Within the graph representation, the terms “jug” and “glass”, presented a similar co-occurrence. Linked to “glass” is the term “water”, which, in turn, organizes two groups of elements: the first, “to drink”, refers to the general actions that can be carried out with water, while the second, “river” describes some characteristics of water in nature. Finally, together with the “river” term, there are also, on the one hand, “bottle” and “bottle cap” with a similar occurrence, both terms referring to elements used to consume water. Also, attached to the term “river” are “fish” and “blue”, which also describe some characteristics of water in nature.

In the children’s data prototypical analysis, the minimum frequency of mention was 3.7, with an evocation range of 2.7. For this analysis, 20 different words were obtained (Table 1). In the central core, we identified the terms: “health”, “drink”, “glass”, “hydration”, “nutrition”, “river”, “jug”, and “water”. Within the contrast zone, the area where elements with low mention frequency are described, but with a low appearance range (first mentioned by some children), are the terms: “refreshing”, “delicious”, “white”, “drink”, and “bottle”. In the first periphery, there are the following words: “drinking water” and “sea”, while in the second periphery, there are the words: “blue”, “bottle cap”, “nature”, “fruit water”, and “transparent”.

The prototypical analysis allows the description of three dimensions: the first, a functional dimension of water, related to health and nutritional aspects (“health”, “nutrition”, and “hydration”); the second, observed in the central core, in the second periphery, and in the contrast zone, describes the instruments used to consume drinking water (“jug”,” bottle”, “glass”, “bottle cap”); and the third, refers to the characteristics of water and its relationship to nature (“drinking water”, “sea”, “blue”, “river”, “nature”, and “transparent”), words that can be found in the four areas of the SR.

### 3.2. Social Representation of Parents of Children from Two Schools in Zapopan, México

With the information provided by the parents, a total of 114 words were obtained from the free listings, of which 59 (51.8%) were similar. In the analysis of the similarity of the SR of the parents, two groups of words were identified (Figure 2). The central element is the term health that organizes the words: “thirst”, “thirst-quencher”, “refreshing”, “fresh”, “purification”, “satisfaction”, “well-being”, and “clean”. In turn, the term “health” relates to the word “hydration”, and this, with “heat”.

The analysis of similarity of the SR of the induction concept, in parents of schoolchildren, represented in the graph, expresses two groupings of the words according to the co-occurrence index, reflecting their consensus regarding the concept of “health”.

Finally, Table 2 describes the prototypical analysis of the SR structure of the schoolchildren’s parents. The minimum range was two and the frequency 5.5. Twelve different words were obtained. At the core, we identified the terms “health” and “thirst”. In the contrast zone are the terms “fresh” and “sugar-free”. Meanwhile, there is the word hydration, in the first periphery zone, and in the second periphery are the terms “clean”, “well-being”, “satisfaction”, “refreshing”, “heat”, and “purification”.

The parents’ data prototypical analysis allows us to describe a functional dimension with the elements: “health”, “thirst”, and “hydration”. On the other hand, there is a dimension regarding the integral well-being that water provides: “well-being”, “satisfaction”, “refreshing”, “purifying”, and “cleaning”. Finally, it highlights a practical dimension, which describes the qualities or elements related to drinking water consumption (“fresh” and “without sugar”).

## 4. Discussion

The objective of the present study was to describe the structure and organization of the SR of “drinking water” in children and parents of two elementary schools from Zapopan, Mexico.

There is an agreement between the results of the similarity analysis and the prototypical analysis, which allows the recognition and description of dimensions that refer to functional, practical, and theoretical aspects. In children and parents, the central element of SR is related to “health”, and, from it, the functional dimension of water emerges. This dimension, also, is linked to aspects of “nutrition” and “hydration”. Some studies carried out in other contexts agree that drinking water is considered healthy by children and parents [24,25,26,27,28,29,30,31], or that it is necessary to prevent disease [28]. This is consistent with what the participants reported: both children and parents consider drinking water to be an essential element in maintaining proper hydration and health.

Our study’s findings evidence that the central core of these SR is focused on the positive effect of drinking water on health, as a concept solidly established. Probably, the study population has been repeatedly exposed to this concept, through individual and collective experiences. For example, the speeches heard by students in natural science classes involve water consumption. Notably, during the school year, the students received topics related to nutrition, including adequate intake of water (and positive effect on health), as an essential aspect to prevent being overweight and obesity by making conscious decisions to improve their feeding [32,33,34]. While keeping this in mind, regarding the thematic concepts of health and water, at the time of conducting the interviews, it was observed that the students consumed sugary drinks instead of plain water. However, the participants’ “drinking water” concept may be constructed considering the socially accepted value of drinking water consumption as a health agent [4]. This same functional dimension of drinking water seems to be identified in parents, as observed in the relationship between health and well-being, aspects that they link to water consumption. In parallel, in parents, there is a functional dimension related to “health” and “hydration”, with the presence of “thirst” and “hot” weather; however, this relationship may be due to the fact that the information collection was carried out during the spring in Mexico (the time of year with the highest heat and the least amount of rainfall). In the present study, this is a particular finding.

On the other hand, in the SR of the children of this study, the terms bottle cap, bottle, glass, and jug were grouped, giving rise to a practical dimension on the instruments used to store and consume drinking water. This dimension suggests that, for children, the instrument that contains drinking water is an important aspect that can influence consumption. Despite having a potable water network, Mexico has safety problems, and, for this reason, tap water is not for consumption. Also, water consumption through drinking fountains is not a culturally accepted practice [35,36]. They are related to the presence of dirt, and they are even considered a source of disease transmission. Specifically, Mexican people believe that drinking fountains are reservoirs of the dengue mosquito (Aedes aegypti), an epidemic that has plagued the region since 2009. Although the hygienic management of drinking fountains must be addressed to eliminate this belief, it would also be useful to promote their use from a sustainability perspective.

The urban population usually consumes drinking water distributed in 20 L jugs that are delivered to the home, or that can be filled out or bought, either in processors or convenience stores. Respectively, Letona, Chacon, Roberto, and Barnoya [37] and Craemer [26] reported that the way food and drinks are presented to children is a crucial factor for their consumption. This aspect can be seen as an opportunity to stimulate drinking water consumption through interventions that offer children instruments such as attractive reusable bottles for drinking water [12].

According to the findings of the children, there is a theoretical dimension about the characteristics of water and its relationship with nature; in other words, children identify the origin of water, as well as the physical characteristics of the water as an element in nature. This aspect is also related to the thematic content of the natural sciences class since it includes the topic of the cycle and natural sources of water. In this regard, it was observed that there are terms related to the natural origin of water within the peripheral elements of children’s SR. This can be an opportunity to stimulate the consumption of drinking water along with caring for the environment, since the use of reusable bottles instead of disposable bottles, can be promoted, in addition to giving value to water as a health vehicle for a healthy environment. This opportunity especially appears when considering the peripheral elements of children’s SR, which are considered elements of change.

Finally, in the contrast zone of the prototypical analysis of adults, there are two terms related to the consumption of drinking water that indicate that it must be “refreshing” and contain no sugar (“without sugar”), indicating that these elements are important since they are associated with healthy habits [38]. So, maybe, the participants have been exposed to marketing campaigns to increase bottled water consumption, which uses these phrases as the main components of their sales messages. In other studies [25,27], these representations have not been reported in adults. However, they do identify how these characteristics of drinking water are also associated with these kinds of habits related to the terms of the central nucleus of SR, for those parents motivated to participate in the study.

The main limitation of this study is the generalization of the results, which may not reflect the SR of the entire schoolchildren population in the Guadalajara metropolitan area, nor of the state, and even less of the country, due to the great diversity of the population. However, this is a characteristic of qualitative research: the search for depth in the knowledge of individuals or groups. According to this, another limitation of this work is the level of analysis, since SR explores the reality of each individual, not of groups.

It would have been more enriching to have informants from different elementary schools in different areas of the metropolitan, urban, and rural areas. However, the particular objective for this study was to explore the context of the participants to design and contextualize an educational intervention aimed at them (phase 2 of the Water Droplet Project).

Another limitation of this study is that it is not possible to generate a theory; however, the findings obtained from this study can infer certain relationships or elements that explore the existing structures between the elements of the SR, given that the vision of each participant is not deepened. The latter makes it possible to address possible groupings of meanings among the participants.

One of the strengths of this work is that, based on its findings, it was possible to contextualize the content of an educational intervention, which would allow it to be relevant to the study population. Although the findings of this study and the content of the derived intervention [39] may not be replicable in other contexts, the methodological model is. This represents another strength of this study: the methodological capacity to be replicated in other contexts, which allows exploring and describing the SRs of drinking water and generating interventions relevant to the research problem.

Although there are different barriers that hinder the consumption of drinking water in children, such as misperceptions, availability, and safety of drinking water [4,28,35,36,40,41], this study only explored the first ones; however, water safety has gained relevance in Mexico [42,43,44,45], particularly in Jalisco [40,41]. Despite the fact that this is a priority issue in public health, it is necessary to continue strengthening strategies to increase water consumption from other perspectives.

## 5. Conclusions

The present study provides a description of the SR of children and parents on “drinking water”. The central element of SR is the term “health”, both in children and parents, which suggests a functional dimension of “drinking water” and its relation to aspects of health, hydration, and nutrition. Nevertheless, in parents, a dimension related to the integral well-being of the population was presented. In addition, in children, a practical dimension of “drinking water” stands out, related to the instruments used for its consumption, such as glasses and bottles.

The results of this research contribute to describing the structure of SR and, therefore, to contextualize and understand the actions of schoolchildren and their parents. That is why the SR of the participants is a starting point to design strategies and interventions for behavior changes, like increasing drinking water consumption in children.

One of the theoretical and methodological contributions of the present study is to show a scientific approach to the SR of the population, with rapid and straightforward information collection strategies that allow a theoretical and structural approach.

However, the coincidence in these SR of the health element, as an element of the central core, for both children and adults, as well as the characteristics and benefits of its consumption, and the importance of containers for its use, were important aspects that were considered to design an educational intervention in this community.

## Figures and Tables

**Figure 1 nutrients-13-01871-f001:**
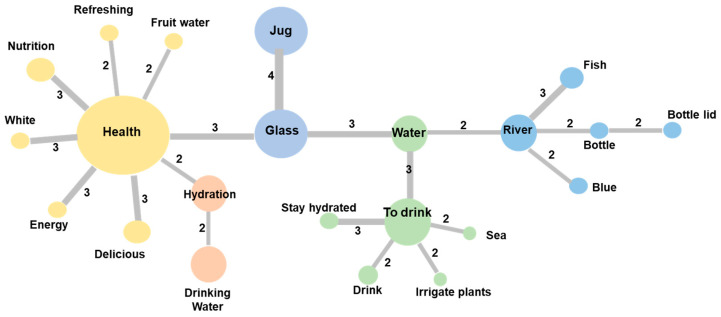
Graphic representation of the similarities of the words related to the social representation of “drinking water”, in schoolchildren from Zapopan, Mexico (*n* = 50). The words presented in the figure are those mentioned concurrently among the participants. The size of the sphere is proportional to the number of times the word was mentioned (the larger the number, the greater the co-occurrence). The number makes reference to the number of times the term was mentioned, while the colors represent groups of semantic proximity.

**Figure 2 nutrients-13-01871-f002:**
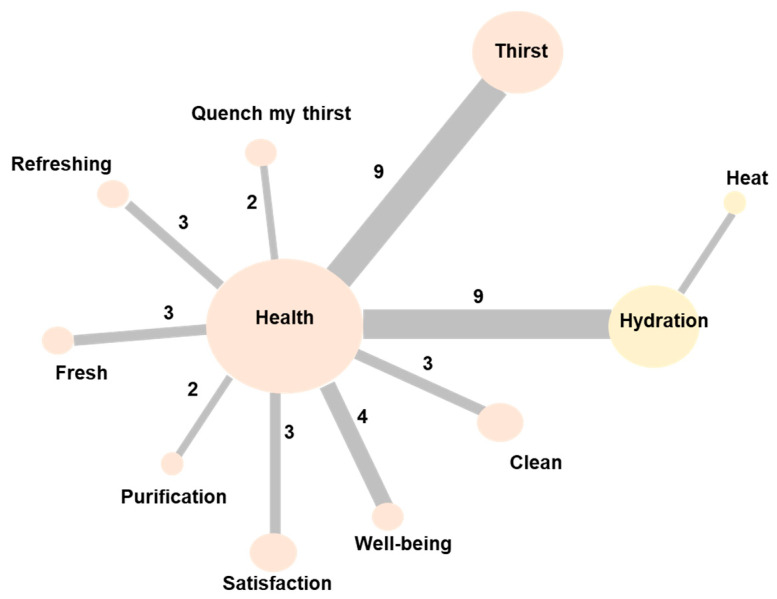
Graphic representation of the similarities of the words related to the social representation of “drinking water”, in parents of schoolchildren from Zapopan, Mexico (*n* = 23). The words presented in the figure are those mentioned concurrently among the participants. The size of the sphere is proportional to the number of times the word was mentioned (the larger the number, the greater the co-occurrence). The number makes reference to the number of times the term was mentioned, while the colors represent groups of semantic proximity.

**Table 1 nutrients-13-01871-t001:** Prototypic analysis of the social representation of “drinking water” in schoolchildren from Zapopan, Mexico (*n* = 50).

	Evocation Rank < 2.7			Evocation Rank ≥ 2.7		
	Central Core			First Peripheral Zone		
Frequency ≥ 3.7	Evoked term	Frequency ^1^	Evocation average ^2^	Evoked term	Frequency ^1^	Evocation average ^2^
	Health Drink Glass Hydration Nutrition River Jug Water	18 8 8 7 6 5 5 5	1.0 2.5 2.2 2.0 2.2 2.4 2.6 2.2	Drinking water Sea	6 4	2.7 4.0
Frequency < 3.7	**Zone of Contrast**			**Second Peripheral Zone**		
	Evoked term	Frequency ^1^	Evocation average ^2^	Evoked term	Frequency ^1^	Evocation average ^2^
	Refreshing Delicious White Drink Bottle	3 3 3 3 3	2.0 2.3 2.0 2.0 2.0	Blue Bottle cap Nature Fruit water Transparent	3 3 3 3 3	7.0 3.3 3.0 3.4 4.0

^1^ Note. Item’s mention frequency in the study population/words frequency in the study population. ^2^ Note. Evocation Rank of the words. The evocation rank indicates the order or position in which the word has been mentioned among the participants. Therefore, the smaller this indicator is, the earlier the word was mentioned. In contrast, a greater number indicates that the word has been mentioned in the last places.

**Table 2 nutrients-13-01871-t002:** Prototypic analysis of the social representation of “drinking water” in parents of children from elementary schools in Zapopan, Mexico (*n* = 23).

	Evocation Rank ≤ 2.7			Evocation Rank > 2.7		
	Central Core			First Peripheral Zone		
Frequency ≥ 5.5	Evoked term	Frequency ^1^	Evocation average ^2^	Evoked term	Frequency ^1^	Evocation average ^2^
	Health Thirst	18 10	2.1 2.5	Hydration	10	2.9
Frequency < 5.5	**Zone of Contrast**			**Second Peripheral Zone**		
	Evoked term	Frequency ^1^	Evocation average ^2^	Evoked term	Frequency ^1^	Evocation average ^2^
	Fresh Sugar-free	4 3	2.5 2.7	Clean Well-Being Satisfaction Refreshing Heat Purification	5 4 3 3 2 2	3.0 4.5 3.0 3.0 4.0 3.0

^1^ Note. Item’s mention frequency in the study population/words frequency in the study population. ^2^ Note. Evocation Rank of the words. The evocation rank indicates the order or position in which the word has been mentioned among the participants. Therefore, the smaller this indicator is, the earlier the word was mentioned. In contrast, a greater number indicates that the word has been mentioned in the last places.

## Data Availability

Data described in the manuscript will be made available upon reasonable request from the corresponding author.

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
