# Peer review of "Social Representations of Drinking Water in Schoolchildren and Parents from Two Schools in Zapopan, Mexico"

_nutrients, 2021, doi:10.3390/nu13061871_

Round 1

Reviewer 1 Report

In my opinion the manuscript entitled „Social representations of drinking water in school children and parents from Zapopan, Mexico” does not make a significant contribution to the issue of drinking water in Mexican children. I want to emphasize that the problem of too low water consumption by Mexican children is well understood, while the results presented in the manuscript do not change the general knowledge about this problem. The problem of water quality and low consumption was presented in such works as:
-    Carriedo et al., 2013. Use of social marketing to increase water consumption among school-age children in Mexico City.
-    Elder et al., 2017. Promotion of water consumption in elementary school children in San Diego, USA and Tlaltizapan, Mexico.
-    Macedo et al., 2019. Safe Water Community Project in Jalisco, Mexico.
-    Smith et al., 2020. Lack of Safe Drinking Water for Lake Chapala Basin Communities in Mexico Inhibits Progress toward Sustainable Development Goals 3 and 6.
You can see that the main focus is now on improving Mexico's water quality. It seems that it would be more legitimate to implement specific programs aimed at increasing children's water consumption, rather than diagnosing the problem in turn, in only one region of the country. It is known that low water consumption is caused by poor water quality, not by reluctance or misperception by children. For this reason, I am not convinced that duplication of existing research is justified.
The issue of low water consumption in Mexico and its determinants is well understood therefore, the presented manuscript does not bring new, valuable knowledge. Intervention programs have been proposed in other articles, barriers and factors facilitating their implementation have been shown, so the presented in manuscript results are not a contribution valuable enough. 
In addition, I have objections to the methodology. It is not known what was the guiding principle behind the selection of schools to participate in the study, if it was targeted selection, it should be explained why this method was chosen.

Author Response

Dear Reviewer:

We appreciate the comments on our paper. They have been instrumental in enriching it and better conveying the contribution it makes regarding previous literature and strengthening points that may not have been sufficiently clear.

In order to better respond to your observations, we responded in to your remarks in greater depth, and we deconstructed them, to make ourselves clearer and respond to each concern individually.

  • In my opinion the manuscript entitled “Social representations of drinking water in school children and parents from Zapopan, Mexico” does not make a significant contribution to the issue of drinking water in Mexican children.

We consider that the work does provide information on the issue of drinking water in Mexico. It is important to point out that this project is part of a mixed methodology study based on Creswell (2007), which had the objective of increasing water consumption and decreasing the consumption of sugar-sweetened beverages. To achieve this objective, we first conducted a qualitative study which allowed us to obtain information about the environment in which the participants of our study live.

This work focuses on knowing or exploring the social elements, beliefs, attitudes, and cultural issues of the participants through the Social Representations Theory (SRT) (Abric, 2001). In other words, the main objective of the work was to emphasize the need to deepen the description and understanding of the study phenomena and the context of the population for the development of culturally appropriate feeding behavior change interventions. As such, the description of these social elements in our article allowed us to contextualize and design the topics for a previously published intervention (Corona Romero et al., 2020).

This is in accordance with what is mentioned by Tumilowicz et al (2015), who points out that it is important to know the population's environment to design interventions appropriate to their characteristics and needs. Several interventions that have focused on increasing drinking water consumption in schoolchildren have been found (Vargas-Garcia et al., 2017), and Mexico has not been the exception. For example, Carriedo et al. (2013) and Elder et al (2014)) designed interventions based on formative research to design their own interventions according to the characteristics of their study population, ensuring their relevance. This characteristic precisely makes this type of design relevant, i.e., the need to implement strategies that bring the use of theory closer to the real needs of each population.

It should be noted that there are no studies that specifically investigate the social representations (SR) of water consumption in children and their parents, and that these representations are the framework for the development of an intervention to increase water consumption, which evidences the originality of our study.

  • I want to emphasize that the problem of too low water consumption by Mexican children is well understood, while the results presented in the manuscript do not change the general knowledge about this problem.

Although the characterization of low water consumption in Mexican children has been clearly exposed, we did not find studies in the western region of the country that analyze this problem from a social perspective; although it is not described in-depth, it provides an approach to the elements. Therefore, this study provides useful information for those who wish to work on this problem, as evidence of the contextualization process, based on the characterization of the SR and the perception of drinking water consumption by those involved. In this sense, this work provides valuable information on the understanding of the phenomenon in western Mexico, specifically in Zapopan, Jalisco.

  • The problem of water quality and low consumption was presented in such works as:

- Carriedo et al., 2013. Use of social marketing to increase water consumption among school-age children in Mexico City.

- Elder et al., 2017. Promotion of water consumption in elementary school children in San Diego, USA and Tlaltizapan, Mexico.

-Macedo et al., 2019. Safe Water Community Project in Jalisco, Mexico.

-Smith et al., 2020. Lack of Safe Drinking Water for Lake Chapala Basin Communities in Mexico Inhibits Progress toward Sustainable Development Goals 3 and 6.

You can see that the main focus is now on improving Mexico's water quality. It seems that it would be more legitimate to implement specific programs aimed at increasing children's water consumption, rather than diagnosing the problem in turn, in only one region of the country. It is known that low water consumption is caused by poor water quality, not by reluctance or misperception by children. For this reason, I am not convinced that duplication of existing research is justified [ … ] Intervention programs have been proposed in other articles, barriers and factors facilitating their implementation have been shown, so the presented in manuscript results are not a contribution valuable enough. 

Our study focuses on evaluating the social representations of school children and parents, providing helpful information for designing and implementing a targeted intervention to increase water consumption in these children. Although not described in this manuscript, another already published paper considered these SRs and contextualized the thematic contents and didactic tools to be more appropriate to the previously studied population (Corona Romero et al., 2020). The manuscript regarding the effect of such intervention built from this work is also in process.

Indeed, our study does not focus on assessing water safety. However, we recognize that several barriers interfere with drinking water consumption and that water safety is a crucial factor in increasing water consumption.

Water contamination in Mexico is a public health problem that has gained recognition in recent years (Arcega-Cabrera et al., 2017; Flor Arcega-Cabrera & Fargher, 2016; Fernández-Macias et al., 2020, Limón-Pacheco et al., 2018). This recognition has been made apparent because of the harmful effects on the population's health (García-Rico et al., 2019; Ramírez-Hernández et al., 2018), especially in the state of Jalisco, in populations near the Chapala lake. Therefore, it is essential to continue exploring and designing interventions to increase water consumption in those environments where water safety may not currently be a priority.

Nevertheless, water safety is not the only barrier to water consumption and is not always a concern in communities where access to water is safer, as in our study. Other barriers that limit drinking water consumption in children have to do with erroneous beliefs, dietary tastes, and availability. For example, Jiménez-Aguilar et al. (2021) and Onufrak et al. (2014)described the perception of Mexican and U.S. children, respectively, regarding water consumption from drinking fountains, which they describe as unhealthy. Hess et al., (2019)for their part, found that U.S. children have the perception that water is unappetizing. In turn, other works have reported incorrect beliefs about beverages and their health attributes. For example, it seems that the sugary drink is considered of natural origin because it indicates that it is made of fruit (Beck et al., 2014; Block et al., 2013; Patel et al., 2014). Also, it has been reported that because it has electrolytes, it has a greater hydration capacity compared to water or that a sugary drink is healthy because it contains vitamins and minerals (Craemer et al., 2013). In addition, the limited availability of drinking water in the child's environment is also a barrier to consumption (Loughridge & Barratt, 2005). This data suggest that it is helpful to perform the first diagnosis on the environment since, according to the context of each population, this may be different and this, determine the success of an intervention for behavioral changes.

Regarding the four articles you mentioned at the beginning of this annotation, we would like to highlight some very relevant points that have undoubtedly contributed to a better understanding and approach to this complex problem in Mexico. We will also point out why we consider that our work contributes to other relevant aspects.

The first two studies Carriedo et al (2013) and Elder et al. (2014) focus on evaluating the effect of interventions that aimed to increase water consumption in schoolchildren. The first was conducted in an area of Mexico City. In contrast, the second study was conducted in an area of Morelos (central Mexico) and San Diego (United States). The design of the interventions of both studies started from formative research (quantitative-qualitative process to support the development of interventions with the idea of adapting them to the culture, geography, and economic aspects (Gittelsohn et al., 2006). This formative research aimed to generate the content of the intervention, contextualizing the constructs of social marketing theory and operant psychology, respectively. These studies provide an effective intervention to increase water consumption and an insight into the need to develop interventions that are theoretically supported and appropriate for each population. Therefore, the information provided by these articles may not be relevant to our population. Although Mexico City, Morelos, and Jalisco are in the same country, different contexts may exist, so it is necessary to describe these environments to identify the similarities and differences between the populations.

In addition to this, the formative research used in the above-mentioned works is a beneficial strategy, but, in our case, we decided to describe the social elements that make up this phenomenon, using SRT from a structuralist perspective. This approach allows us to describe the social knowledge about drinking water consumption in the study population; by using the free listing technique (a tool used for data collection), we can access these elements quickly and spontaneously in the participants (school children and mothers). One of the strengths of this work is that precisely these findings are presented, which in comparison by the previous works Carriedo et al (2013  and Elder et al. (2014), aim to show the results of their respective interventions.

This is one of the main differences between the formative research employed by Carriedo et al. (2013) and Elder et al. (2014), as they focus on complementing the constructs of the theories used during their studies through the findings; whereas, for this work, the use of SRT was proposed to explore a phenomenon and, based on the findings, contextualize an intervention.

The other two articles mentioned have to do with a severe problem related to health and water in the area relatively close to Zapopan, which is Ribera de Chapala (60 km away). For more than two decades, the problem of the low quality of water in the populations bordering the shore of Lake Chapala has been analyzed, as well as its impact on the prevalence and incidence of diarrhea and chronic kidney disease in the population.  

The article by Macedo et al. (2019) presents the research protocol that consists of the first phase of observation and the second phase of implementation, where an intervention will be carried out to improve water quality, sanitation, and hygiene. Specifically, this work does not consider other aspects related to drinking water consumption, in addition to taking the general population as its sampling frame. Regarding the article by Smith et al. (2020), the objective of this work is to describe the problem of water contamination in the Chapala riverbank and surrounding municipalities through a mixed methodology study, which includes a geospatial localization method that allows researchers to determine patterns with the houses that presented water contamination. The results describe the levels of arsenic and fecal contamination (E. coli) in the main primary and secondary water sources within the studied population. In addition, it is evident that 30 to 60% of the population consumes this contaminated drinking water, regardless of its source (jug or tap water).

As can be seen, these articles address relevant problems, but they were not the focus of our work, since in the metropolitan urban area of Guadalajara, which includes the municipality of Zapopan, the water from the Chapala riverbank is treated to make it suitable for human consumption.

  • The issue of low water consumption in Mexico and its determinants is well understood therefore, the presented manuscript does not bring new, valuable knowledge.

As mentioned in point one, the objective of this work is not to learn about the issue of low consumption of drinking water in children, but rather to identify which aspects of their beliefs and social representations should be considered when designing an intervention. This study is a frame for the implementation of that intervention and the evaluation of its effect on water consumption. To date, we have not been able to find in the literature any work similar to the one we present that describes SRs on drinking water consumption in children and their parents.

Moreover, after more than a year of virtual classes for primary school students, this SR characterizes a pre-pandemic reality, which may never be experienced in the same way again, and would be described for posterity with this document.

We thank you for your valuable comments about the above mentioned points. Based on them, we have included some of this especially important research concerns that involved, our region (lines 331-337).

  • In addition, I have objections to the methodology. It is not known what was the guiding principle behind the selection of schools to participate in the study, if it was targeted selection, it should be explained why this method was chosen.

Although qualitative research does not seek for generalization of data and convenience sampling is often selected, we decided, we decided to make this description of the random selection of schools because it justified the representativeness of the intervention we wanted to construct afterwards.

Thus, the selection of the primary schools was made randomly. The inclusion criteria were the following: schools belonging to the Ministry of Public Education of the state of Jalisco (SEP), located in the Metropolitan Area of Guadalajara, mixed girls and boys’ schools, and a medium socioeconomic level. Approximately, ten elementary schools were selected that met these characteristics. Unfortunately, only one of the ten schools obtained a favorable response from the authorities to carry out the study. Concerning the private elementary school, it was decided to select schools close to the area where the public school was selected to characterize these two contexts, in the same geographic area and of similar socioeconomic level. A total of six schools were located that met these characteristics; however, only one agreed to participate. On the other hand, time, human and economic resources were limited.

A convenience sampling was used to select the schoolchildren; we included students from third to fifth grade who decided to participate voluntarily, who had given their verbal consent, and whose parents had signed the voluntary informed consent form that was sent to them at the beginning of the study. The parents/guardians of the participating children were also invited to participate through an information package sent to them though their children that contained an invitation to the project, information regarding the study, an informed consent letter, the free list format (a technique used for data collection), and the instructions for its completion. Parents who agreed to participate and who returned the signed informed consent form were included in the study.

We thank you for your valuable comments. Based on them, we have included these topics in the document for a better understanding (lines 93-101).

References

Abric, J. C. (2001). Prácticas sociales y representaciones. In J. Dacosta Chavel & D. Flores Palacios (Eds.), Practicas sociales y representación (Primera ed, Issue 47). Ediciones Coyoacán SA de CV.

Arcega-Cabrera, F, Fargher, L. F., Oceguera-Vargas, I., Noreña-Barroso, E., Yánez-Estrada, L., Alvarado, J., González, L., Moo-Puc, R., Pérez-Herrera, N., Quesadas-Rojas, M., & Pérez-Medina, S. (2017). Water consumption as source of arsenic, chromium, and mercury in children living in rural Yucatan, Mexico: Blood and urine levels. Bulletin of Environmental Contamination and Toxicology, 99(4), 452–459. https://doi.org/10.1007/s00128-017-2147-x

Arcega-Cabrera, Flor, & Fargher, L. F. (2016). Education, fish consumption, well water, chicken coops, and cooking fires: Using biogeochemistry and ethnography to study exposure of children from Yucatan, Mexico to metals and arsenic. Science of The Total Environment, 568, 75–82. https://doi.org/https://doi.org/10.1016/j.scitotenv.2016.05.209

Beck, A. L., Takayama, J. I., Halpern-Felsher, B., Badiner, N., & Barker, J. (2014). Understanding how latino parents choose beverages to serve to infants and toddlers. Maternal and Child Health Journal, 18(6), 1308–1315. https://doi.org/10.1007/s10995-013-1364-0

Block, J., Gillman, M., Linakis, S., & Goldman, R. (2013). “If it tastes good, I’m drinking it”: Qualitative study of beverage consumption among college students. Journal of Adolescent Health, 52, 702–706.

Carriedo, A., Bonvecchio, A., López, N., Morales, M., Mena, C., Théodore, F. L., & Irizarry, L. (2013). Uso del mercadeo social para aumentar el consumo de agua en escolares de la Ciudad de México. Salud Pública de México, 55(3), 388–396.

Corona Romero, A. M., Bernal Orozco, M. F., & Vizmanos Lamotte, B. (2020). Diseño de una intervención educativa para aumentar el consumo de agua en niños escolares de Zapopan, México. Revista de Educación y Desarrollo, 52(1).

Craemer, M. De, Decker, E. De, Bourdeaudhuij, I. De, Deforche, B., Vereecken, C., Duvinage, K., Grammatikaki, E., Iotova, V., Fernández-alvira, J. M., & Zych, K. (2013). Physical activity and beverage consumption in preschoolers : focus groups with parents and teachers. Biomed Central Public Health, 278(13), 1–13.

Creswell, J. W. (2007). Qualitative inquiry and research design: Choosing among five traditions. In Qualitative Health Research (Vol. 9, Issue 5, p. 403). https://doi.org/10.1111/1467-9299.00177

Elder, J., Holub, C., Arredondo, E., Sánchez-Romero, L. M., Moreno-Saracho, J., Barquera, S., & Rivera, J. (2014). Promotion of water consumption in elementary school children in San Diego, USA and Tlaltizapan, Mexico. Salud Pública de México, 56(2), S148–S156.

Fernández-Macias, J. C., Ochoa-Martínez, Á. C., Orta-García, S. T., Varela-Silva, J. A., & Pérez-Maldonado, I. N. (2020). Probabilistic human health risk assessment associated with fluoride and arsenic co-occurrence in drinking water from the metropolitan area of San Luis Potosí, Mexico. Environmental Monitoring and Assessment, 192(11), 712. https://doi.org/10.1007/s10661-020-08675-7

García-Rico, L., Meza-Figueroa, D., Jay Gandolfi, A., del Rivero, C. I., Martínez-Cinco, M. A., & Meza-Montenegro, M. M. (2019). Health risk assessment and urinary excretion of children exposed to arsenic through drinking water and soils in Sonora, Mexico. Biological Trace Element Research, 187(1), 9–21. https://doi.org/10.1007/s12011-018-1347-5

Gittelsohn, J., Steckler, A., Johnson, C. C., Pratt, C., Grieser, M., Pickrel, J., Stone, E. J., Conway, T., Coombs, D., & Staten, L. K. (2006). Formative research in school and community-based health programs and studies: “state of the art” and the TAAG approach. Health Education & Behavior : The Official Publication of the Society for Public Health Education, 33(1), 25–39. https://doi.org/10.1177/1090198105282412

Hess, J. M., Lilo, E. A., Cruz, T. H., & Davis, S. M. (2019). Perceptions of water and sugar-sweetened beverage consumption habits among teens, parents and teachers in the rural south-western USA. Public Health Nutrition, 22(8), 1376–1387. https://doi.org/10.1017/S1368980019000272

Jiménez-Aguilar, A., Muñoz-Espinosa, A., Rodríguez-Ramírez, S., Maya-Hernández, C., Gómez-Humarán, I. M., Uribe-Carvajal, R., Salazar-Coronel, A., Sachse-Aguilera, M., Veliz, P., & Shamah-Levy, T. (2021). Consumo de agua, bebidas azucaradas y uso de bebederos en secundarias del Programa Nacional de Bebederos Escolares de la Ciudad de México. Salud Publica de Mexico, 63(1), 68–78. https://doi.org/10.21149/11023

Limón-Pacheco, J. H., Jiménez-Córdova, M. I., Cárdenas-González, M., Sánchez Retana, I. M., Gonsebatt, M. E., & Del Razo, L. M. (2018). Potential Co-exposure to arsenic and fluoride and biomonitoring equivalents for Mexican children. Annals of Global Health, 84(2), 257–273. https://doi.org/10.29024/aogh.913

Loughridge, J. L., & Barratt, J. (2005). Does the provision of cooled filtered water in secondary school cafeterias increase water drinking and decrease the purchase of soft drinks? Journal of Human Nutrition and Dietetics, 18, 281–286.

Macedo, E., Rocco, M. V., Mehta, R., & Garcia-Garcia, G. (2019). Safe Water Community Project in Jalisco, Mexico. Annals of Nutrition and Metabolism, 74(Suppl3), 51–56. https://doi.org/10.1159/000500346

Onufrak, S. J., Park, S., Sharkey, J. R., Merlo, C., Dean, W. R., & Sherry, B. (2014). Perceptions of tap water and school water fountains and association with intake of plain water and sugar-sweetened beverages. The Journal of School Health, 84(3), 195–204. https://doi.org/10.1111/josh.12138

Patel, A., Bogart, L., Klein, D., Cowgill, B., Uyeda, K., & Hawes-Dawson, J. (2014). Middle school student attitudes about school drinking fountains and water intake. Academic Pediatrics, 14, 471–477.

Ramírez-Hernández, H., Perera-Rios, J., May-Euán, F., Uicab-Pool, G., Peniche-Lara, G., & Pérez-Herrera, N. (2018). Environmental risks and children’s health in a Mayan community from southeast of Mexico. Annals of Global Health, 84(2), 292–299. https://doi.org/10.29024/aogh.917

Smith, C. D., Jackson, K., Peters, H., & Lima, S. H. (2020). Lack of safe drinking water for lake chapala basin communities in Mexico inhibits progress toward sustainable development goals 3 and 6. International Journal of Environmental Research and Public Health, 17(22), 1–12. https://doi.org/10.3390/ijerph17228328

Tumilowicz, A., Neufeld, L. M., & Pelto, G. H. (2015). Using ethnography in implementation research to improve nutrition interventions in populations. Maternal and Child Nutrition, 11, 55–72. https://doi.org/10.1111/mcn.12246

Vargas-Garcia, E. J., Evans, C. E. L., Prestwich, A., Sykes-Muskett, B. J., Hooson, J., & Cade, J. E. (2017). Interventions to reduce consumption of sugar-sweetened beverages or increase water intake: evidence from a systematic review and meta-analysis. Obesity Reviews. https://doi.org/10.1111/obr.12580

Reviewer 2 Report

Water consumption is very low in Mexico. The aim of this study was to describe the social representations (SR) of drinking water in school-children (n=50) and parents (n=23) of two schools in a large city in Mexico, Zapopan. Associative free listing was used as information gathering technique. A similarity analysis was performed using the co-occurrence index. Prototypical analysis was performed to explore the structure of the SR. Three dimensions were described in the children's SR: a functional dimension related to health and nutrition, a practical dimension that describes the instruments used for its consumption, and a theoretical dimension that specifies the characteristics of water and its relationship with nature. In the parents’ SR a functional dimension was also found; another dimension was described regarding the integral well-being that drinking water provides, and a practical dimension that describes the features related to its consumption. This study gives some insights in the associations school children and parents have with water, which can be useful in designing public health campaigns for example to increase water consumption.

The manuscript provides some interesting findings, but there are several issues that need to be addressed to strengthen this report.

Typo error in first sentence abstract: reciently should be recently. Also, Line 48: according to this problematic, should be according to this problem. Please spellcheck the complete manuscript.

Major points:

Title reads:  Social representations of drinking water in school children and 2 parents from Zapopan. However, as only two schools are included in this analysis, this title is a bit misleading as Zapopan is a huge city. Suggest to include ‘two schools’ in the title.

Par 2.2

Why were these two schools chosen, what was the reasoning for choosing one private and one public from this specific city? Would it not have been better to include a few more schools to get a better spread over the city? As Zapopan is a large city. Also, it would have been great if schools outside the city would have been included, as there might be differences in people from rural and urban areas for example. Or was this just a pilot study? I understand it’s a qualitative study and the authors try to address this as a limitation of the paper. But even for qualitative studies you try to reach at least different ‘types’ of people to get different views and you are trying to get to a saturation level. And I have the concern that this sample is rather limited and hence will be less informative for the reader.

Figure 1: text is really hard to see as the background is black. Also, a lot of words can’t be read because of this.

For the tables, how is the ‘frequency’ cut-off choses? Is it the median? This needs to be described either in the methods or in the results section.

And what about the ‘range’ cut-off? Why is that similar for parents and children, but the ‘frequency’ one isn’t?

Discussion

First paragraph: the authors state “From this, it is possible to generate theoretical approaches and structural assumptions that allow us to know and explain the concept of drinking water in the population” Is this really possible? As the sample is only based on two schools? Suggest to rephrase this sentence.

Lines 249-260: include references to support the facts/statements made about water fountains.

Minor points

Line 48-49: a water recommendation is mentioned here, but this article that is references here only shows that of the total fluid intake, only 30% is water for children from Mexico. That is something else then 2 30% of recommended intake’. Please rephrase

Lines 96-97: the authors stated: “The selection of the study population was carried out through non-probability sampling, in order to focus in-depth on the information that the participants can and choose to share.” Could the authors please elaborate on this please, as I don’t see the connection between these two?

Line 98: please include the age-range of the children.

End of par 2.3: the packages for parents, were these send by post? It says ‘a package was previously send..’ What does the word ‘previously’ means in this context? Was is send to parents before children were interviewed? Or as in ‘this was done in the past’? Please just describe a bit more what is meant here as the word ‘previously’ does not seem appropriate here.

Author Response

Dear Reviewer:

We appreciate the comments on our paper. In order to better respond to your observations, we responded in to your remarks in greater depth, and we deconstructed them, to make ourselves clearer and respond to each concern individually.

Water consumption is very low in Mexico. The aim of this study was to describe the social representations (SR) of drinking water in school-children (n=50) and parents (n=23) of two schools in a large city in Mexico, Zapopan. Associative free listing was used as information gathering technique. A similarity analysis was performed using the co-occurrence index. Prototypical analysis was performed to explore the structure of the SR. Three dimensions were described in the children's SR: a functional dimension related to health and nutrition, a practical dimension that describes the instruments used for its consumption, and a theoretical dimension that specifies the characteristics of water and its relationship with nature. In the parents’ SR a functional dimension was also found; another dimension was described regarding the integral well-being that drinking water provides, and a practical dimension that describes the features related to its consumption. This study gives some insights in the associations school children and parents have with water, which can be useful in designing public health campaigns for example to increase water consumption.

The manuscript provides some interesting findings, but there are several issues that need to be addressed to strengthen this report.

1.- Typo error in first sentence abstract: reciently should be recently. Also, Line 48: according to this problematic, should be according to this problem. Please spellcheck the complete manuscript.

Thanks for your review; we have made the pertinent changes within the manuscript. Also, the text has been reviewed by an expert.

Major points:

2.- Title reads:  Social representations of drinking water in school children and parents from Zapopan. However, as only two schools are included in this analysis, this title is a bit misleading as Zapopan is a huge city. Suggest to include ‘two schools’ in the title.

We thank you for your suggestion regarding the title of the manuscript. We changed the title and incorporated this. The proposed title is: Social representations of drinking water in schoolchildren and parents from two schools in Zapopan.

  1. Par 2.2 Why were these two schools chosen, what was the reasoning for choosing one private and one public from this specific city? Would it not have been better to include a few more schools to get a better spread over the city? As Zapopan is a large city. Also, it would have been great if schools outside the city would have been included, as there might be differences in people from rural and urban areas for example. Or was this just a pilot study? I understand it’s a qualitative study and the authors try to address this as a limitation of the paper. But even for qualitative studies you try to reach at least different ‘types’ of people to get different views and you are trying to get to a saturation level. And I have the concern that this sample is rather limited and hence will be less informative for the reader.

We thank you for your comments.

It is important to mention that this study, "Social representations: drinking water consumption for children and parents of two schools in Zapopan," is part of a macro project which is framed in a mixed methodology, with an exploratory sequential design (Creswell 2007)). In this study, the first stage consists of a qualitative study, which for specific purposes of this project, we used the Theory of Social Representations (Moscovici 1979) with a structuralist approach (Abric 2001). The objective of this phase was to explore the social phenomenon of natural water consumption in a specific context (elementary schools in Zapopan, a city in the urban area of Guadalajara). The findings obtained in this study were used to contextualize the design of an educational intervention that was implemented in those same elementary schools (lines 83-87).

Therefore, we initially obtained information/data that would allow us to know and/or explore the environment in which our participants develop and their social representations regarding this topic. Because of this, we only included students and parents from these two elementary schools.

The selection of the elementary schools was made through random sampling. Approximately ten elementary schools belonging to the Secretary of Public Education of the state of Jalisco (SEP) were selected. The inclusion criteria were that the schools belonged to the SEP in the state of Jalisco, that they were located within the metropolitan area of Guadalajara (that includes Zapopan), and that their socio-economic level was medium. Unfortunately, only one of the ten schools obtained a favorable response from the authorities to carry out the study project. With respect to the private elementary school, it was decided to choose one close to the area where the public one was selected in order to characterize these two contexts in the same geographical area and with a similar socio-economic level (lines 93-101).

It is true that in order to know in depth a social phenomenon within a population, it is necessary to have a more heterogeneous population. It would have been more enriching to have informants from different primary schools in the metropolitan area of Guadalajara, as well as from urban and rural areas, given that it is known that the situation of consumption and availability of drinking water is different in each context (lines 314-330). However, our particular objective for this study was to explore the context of our participants in order to design and contextualize an educational intervention aimed at them, as we believe that it could have a greater positive impact on water consumption, as indicated in other studies (Tumilowicz, Neufeld, and Pelto 2015).

However, we recognize that when formulating an intervention with these findings, it is also a major limitation, as the data and the design of the intervention may not be replicable for other contexts.

4.- Figure 1: text is really hard to see as the background is black. Also, a lot of words can’t be read because of this.

We appreciate this observation. We did not realize that the colors used in the image might be difficult to see. We have redesigned the images (Figure. 1 and Figure 2) so that they can be more legible and understandable to the reader. Thank you.

Figure 1. Graphic representation of the similarities of the SR words of school children in Zapopan, Mexico (n=50)

Figure 2. Graphical representation of word similarities of SRs of parents of school children from Zapopan, Mexico (n=23). 

5.- For the tables, how is the ‘frequency’ cut-off choses? Is it the median? This needs to be described either in the methods or in the results section.

And what about the ‘range’ cut-off? Why is that similar for parents and children, but the ‘frequency’ one isn’t?

Thank you for your comment. The frequency of mention is a criterion that allows us to know the number of times the word has been mentioned by our participants (Abric 2001). Therefore, it was decided to establish as a criterion of "frequency of mention of the words" that at least "2 times". It was necessary the same word to be mentioned at least two times among the participants, considering that as a coincidence in the mention.

According to the nature of the data and for its analysis, it is not possible to use measures of central tendency. According to the prototypical analysis, we use two criteria, in which we weigh: 1) the order of mention given by the participant when mentioning the word and 2) the number of times the word has been mentioned.

Regarding the rank, initially, we would like to clarify that there was an error in the translation "range" is incorrect; the correct form of the rank is "rank," which refers to the order of mention of the words (criterion 1, previously mentioned). This indicator can vary between 1 and 5 (the number of words that the participant has been asked to mention during the application of the free listing survey) since it responds to the order in which the word has been mentioned; therefore, this is usually more stable. In general, this cut-off ranges between 2.5 because there are only five mentions; however, this can vary if the participant provides a greater or lesser number of words. While the frequency can range from 1 to the number of participants, this cut-off point is less stable because it depends on the population. In the case of our study, the number of participants in the populations is different; in the children, we obtained responses from 50 school children, while only 23 mothers, mainly of school children, participated.

We really appreciate this observation. Undoubtedly, adding more information to the manuscript will be of greater support for the correct understanding of the data and the results (lines 140-150). Also, within tables 1 and 2, the term range was modified by the term evocation rank. In addition, footnote number 2 was also modified.

Discussion

6.- First paragraph: the authors state “From this, it is possible to generate theoretical approaches and structural assumptions that allow us to know and explain the concept of drinking water in the population” Is this really possible? As the sample is only based on two schools? Suggest to rephrase this sentence.

While it is true that according to the findings we have obtained from this study, we can infer certain relationships or elements that explore the existing structures between the elements mentioned, these data do not allow us to generate a theory since the vision of each participant is not deepened. In addition, the participants only belong to a particular population (two nearby schools, one private and one public), which may not be representative or relevant for the entire school population of Zapopan, in general, but logically for our participants from the schools on which we defined to focus the intervention.

This situation only allows us to address possible groupings of meanings among participants (this is explained in the lines 324-330).

7.- Lines 249-260: include references to support the facts/statements made about water fountains.

Thanks to your comments, we have been able to include the references that support our statements (lines 2713-274).

Minor points

8.- Lines 48-49: a water recommendation is mentioned here, but this article that is references here only shows that of the total fluid intake, only 30% is water for children from Mexico. That is something else then 30% of recommended intake’. Please rephrase

Thank you for the comment. Changes were made (lines 48-49) and now the information is clearly expressed.

9.- Lines 96-97: the authors stated: “The selection of the study population was carried out through non-probability sampling, in order to focus in-depth on the information that the participants can and choose to share.” Could the authors please elaborate on this please, as I don’t see the connection between these two?

Thanks to your comments, we have been able to reformulate this section.

The selection of participants, both children, and parents, was through convenience sampling (Teddlie and Yu 2007). The information is available on the following lines: 103-105.

10.- Line 98: please include the age-range of the children.

Thank you for your comment. This information has been added on lines 14-15.

11.- End of par 2.3: the packages for parents, were these send by post? It says ‘a package was previously send..’ What does the word ‘previously’ means in this context? Was is send to parents before children were interviewed? Or as in ‘this was done in the past’? Please just describe a bit more what is meant here as the word ‘previously’ does not seem appropriate here.

Thank you for your comment. When the authorities of both public and private elementary schools allowed us to carry out the intervention, as a first step, we contacted the parents of both schools. This was done on the basis of a packet (booklet) that included a letter of invitation to participate in the study, procedures, and activities to be carried out within the school in which the children would be participating. In addition, it included the letter of informed consent for their participation (in duplicate), the format of the free lists with instructions to be answered by the parents (one or both).

Once parental authorization was obtained, during school hours, each child was invited to participate. If we obtained the verbal assent of the child so that their participation was completely voluntary, we assumed that they did want to voluntarily accept, and then, we applied the questions from the free list, and the interviewer wrote down the answers in each case (lines 108-112).

Thank you for your valuable comments to strengthen our article and ensure its reproducibility and applicability.

References:

Abric, Jean Claude. 2001. Prácticas Sociales y Representaciones. Edited by José Dacosta Chavel and Daniela Flores Palacios. Practicas Sociales y Representación. Primera ed. Ciudad de México, México: Ediciones Coyoacán SA de CV.

Creswell, John W. 2007. “Qualitative Inquiry and Research Design: Choosing among Five Traditions.” Qualitative Health Research. https://doi.org/10.1111/1467-9299.00177.

Moscovici, Serge. 1979. El Psicoanálisis, Su Imagen y Su Público. Edited by Finetti NM. 2da edició. Buenos Aires, Argentina: Huemul.

Teddlie, Charles, and Fen Yu. 2007. “Mixed Methods Sampling: A Tipology with Examples.” Journal of Mixed Methods Research 1 (1): 77–100.

Tumilowicz, Alison, Lynnette M. Neufeld, and Gretel H. Pelto. 2015. “Using Ethnography in Implementation Research to Improve Nutrition Interventions in Populations.” Maternal and Child Nutrition 11: 55–72. https://doi.org/10.1111/mcn.12246.

Round 2

Reviewer 1 Report

Authors revised the manuscript in line with the comments.

Reviewer 2 Report

In my opinion, the rebuttal is clear, all comments are carefully and thoughtfully addressed and changes are clearly highlighted by the author, which makes reviewing the revision an easy job, thanks! I have no further comments.